# Integrative multiplatform molecular profiling of benign prostatic hyperplasia identifies distinct subtypes

Deli Liu [1,2,3,4], Jonathan E. Shoag[1,2], Daniel Poliak[5], Ramy S. Goueli[2], Vaishali Ravikumar[2], David Redmond[6], Aram Vosoughi[7], Jacqueline Fontugne[4,7], Heng Pan[4], Daniel Lee[2], Domonique Thomas[2], Keyan Salari [8], Zongwei Wang[9], Alessandro Romanel [10], Alexis Te[2], Richard Lee[2], Bilal Chughtai[2], Aria F. Olumi[9], Juan Miguel Mosquera[3,4,7], Francesca Demichelis[10], Olivier Elemento[3,4], Mark A. Rubin [4,11], Andrea Sboner[1,3,4,7,12 ✉] & Christopher E. Barbieri [1,2,4,12 ✉]

Benign prostatic hyperplasia (BPH), a nonmalignant enlargement of the prostate, is among the most common diseases affecting aging men, but the underlying molecular features remain poorly understood, and therapeutic options are limited. Here we employ a comprehensive molecular investigation of BPH, including genomic, transcriptomic and epigenetic profiling. We find no evidence of neoplastic features in BPH: no evidence of driver genomic alterations, including low coding mutation rates, mutational signatures consistent with aging tissues, minimal copy number alterations, and no genomic rearrangements. At the epigenetic level, global hypermethylation is the dominant process. Integrating transcriptional and methylation signatures identifies two BPH subgroups with distinct clinical features and signaling pathways, validated in two independent cohorts. Finally, mTOR inhibitors emerge as a potential subtype-specific therapeutic option, and men exposed to mTOR inhibitors show a significant decrease in prostate size. We conclude that BPH consists of distinct molecular subgroups, with potential for subtype-specific precision therapy.

[1] Sandra and Edward Meyer Cancer Center, Weill Cornell Medicine, New York, NY, USA. [2] Department of Urology, Weill Cornell Medicine, New York, NY, USA. [3] HRH Prince Alwaleed Bin Talal Bin Abdulaziz Alsaud Institute for Computational Biomedicine, Weill Cornell Medical College, New York, NY, USA. [4] Englander Institute for Precision Medicine of Weill Cornell Medicine and NewYork-Presbyterian Hospital, New York, NY, USA. [5] Department of Radiology, Weill Cornell Medicine, New York, NY, USA. [6] Department of Medicine, Weill Cornell Medicine, New York, NY, USA. [7] Department of Pathology and Laboratory Medicine, Weill Cornell Medical College, New York, NY, USA. [8] Department of Urology, Massachusetts General Hospital, Harvard Medical School, Boston, MA, USA. [9] Beth Israel Deaconess Medical Center, Harvard Medical School, Boston, MA, USA. [10] Department of Cellular, Computational and Integrative Biology (CIBIO), Trento, Italy. [11] Department of BioMedical Research, University of Bern and Inselspital, Bern, Switzerland. [12] These authors jointly supervised this work: Andrea Sboner, Christopher E. Barbieri. ✉email: ans2077@med.cornell.edu; chb9074@med.cornell.edu

Benign prostatic hyperplasia (BPH) is a common disease, affecting nearly all men as they age[1–5]. BPH frequently results in bladder outlet obstruction with concomitant lower urinary tract symptoms or infections, and more rarely bladder decompensation and renal failure[3,6,7]. The prevalence of BPH increases with age, with BPH symptoms reported by roughly 80% of men at age 70–79[1–4,7,8]. Approved medical therapies for BPH are limited to alpha-blockers, 5-alpha reductase inhibitors, and *PDE5* inhibitors[8,9]. However, many patients fail medical therapies, and require surgical intervention[10]. Histologically, BPH is characterized as the overgrowth of stromal and epithelial cells, and it occurs in the transition zone of the prostate[1]. Currently, many BPH studies have focused on risk factors of BPH[11–13], while the underlying molecular features of BPH remain under-studied[3,9,14–16] and molecular data is relatively scarce[17,18]. Moreover, BPH has been described as the most common benign tumor in men, and is commonly referred to as an adenoma, but unlike many malignant[19] and benign neoplasms[20–22], it is unknown whether BPH is a neoplastic process[3,7,15–17]. Genomic driver alterations are identifiable in many benign neoplasms; for instance, uterine leiomyomas harbor recurrent *MED12* mutations as well as complex chromosomal rearrangements[23,24], and profiling of hepatocellular adenomas has revealed multiple recurrent mutations[22]. In this study, we perform a comprehensive investigation of 18 BPH cases via next-generation sequencing technology. We selected samples from patients with very large prostates (top 1 percentile and greater than 100cc), based on the rationale that these extreme outliers were more likely to harbor biologically informative events[25].

## Results

### Genomic alterations and mutational signatures in BPH.
To define the landscape of genomic alterations in BPH, we performed whole genome sequencing (WGS), whole exome sequencing (WES) and SNP arrays on 18 BPH cases and matched controls (Fig. 1a, Supplementary Table 1 and Supplementary Fig. 1). The number of somatic coding mutations (SNV) ranged from 0.1 to 1 per megabase (Mb) (Supplementary Table 2). As compared to neoplastic diseases (benign and malignant)[20–22], BPH samples harbored fewer SNVs (Fig. 1b), and there were no recurrent SNVs to suggest driver alterations. To understand underlying mutational processes, we examined mutational signatures[26] across all BPH cases, and found BPH was highly associated with mutation signature 1[26], which included C > T substitutions at NpCpG trinucleotides (Fig. 1c, d). This signature has been shown to correlate with age[26], consistent with the age-related onset of BPH[1–4,7]. Moreover, BPH samples harbored minimal copy number alterations, and the fraction of altered genome was far lower than seen in primary prostate cancer[19] and other neoplastic diseases (Fig. 1e, f, Supplementary Tables 4 and 5). Also unlike primary prostate cancer[19], analyses of structural variants in WGS revealed no genomic rearrangements in BPH (Fig. 1g and Supplementary Fig. 2). Finally, we explored the possibility of subclonal prostate-specific SNVs occurring at minimal VAF, with direct examination of the reads showing no evidence of these, even at the lowest detectable frequencies (Supplementary Figs. 3 and 4, Supplementary Table 6). Together, these data show no evidence of driver genomic alterations in BPH, inconsistent with a neoplastic disease process.

### Transcriptional landscape of BPH.
We next examined the transcriptional landscape of BPH using RNA-seq. Because BPH, by its very nature often has no adjacent normal tissue, we compared the gene expression profiles from BPH samples with histologically normal transition zone tissue sampled from age-matched controls (Fig. 2a, Supplementary Tables 7 and 8). We identified a BPH transcriptional signature that included 392 differentially expressed genes between BPH and control samples (Fig. 2b and Supplementary Table 9). When compared to control samples from the normal peripheral zone[27–29], this transcriptional signature was BPH specific, and not specific to transition zone tissue (Supplementary Fig. 6). We next validated this BPH transcriptional signature using two independent study cohorts[18,30], and again found reliable clustering of BPH samples (Fig. 2c, d) with similar upregulation of *BMP5* identified (Supplementary Table 9). Having defined and validated a robust set of genes altered in BPH, we explored the signaling pathways deregulated using gene set enrichment analysis (GSEA)[31] (Fig. 2e). Interestingly, multiple signatures related to inactivation of KRAS signaling were observed in our dataset, with concordance in an independent cohort (Fig. 2e), and again inconsistent with a neoplastic process. In addition, we observed AR signaling downregulated in BPH (Fig. 2f and Supplementary Fig. 7), consistent with previous findings that AR signaling disruption correlated with prostate inflammation and BPH pathogenesis[32,33]. However, as is common practice for BPH, all patients were exposed to medications affecting AR activity prior to surgery (5-alpha reductase inhibitors), making it unclear whether AR target gene changes were due to intrinsic properties of BPH or prior therapy.

### DNA methylation landscape of BPH.
Next, we investigated the epigenetic landscape of BPH by defining the DNA methylation profile of 18 BPH samples and 5 controls from normal transition zone tissue using ERRBS (Enhanced Reduced Representation Bisulfite Sequencing). We identified 92,046 hypermethylated CpGs and 10,117 hypomethylated CpGs across different genomic regions in BPH, with hypermethylation being the dominant signal across all genomic regions, even when controlling for bias of CpG-rich loci (Fig. 2g, h and Supplementary Fig. 8). We defined a methylation signature for BPH that included 696 differentially methylated CpGs in promoter regions (Fig. 2i and Supplementary Table 10). Consistent with DNA methylation as a major mechanism of transcriptional control in BPH, we found negative correlation between promoter methylation and gene expression (Fig. 2j). For instance, *HOXD1* was both underexpressed and hypermethylated at the promoter in BPH specimens, consistent with the downregulation of AR signaling pathway found in BPH[34,35] (Fig. 2f and Supplementary Fig. 7).

### Molecular profiling identifies two distinct BPH subgroups.
Identifying distinct molecular subtypes in human disease has provided insight into important biological and clinical phenomena. We therefore performed integrative analysis using transcriptional and methylation profiling, and identified two distinct BPH subtypes (Fig. 3a and Supplementary Tables 9 and 10), supporting robust biologically distinct subgroups across different data types. To validate distinct subtypes in BPH, we tested our signature via k-means clustering in two independent cohorts[17,18], and identified nearly identical subgroups (Fig. 3c, e and Supplementary Table 11), further supporting the robustness of these subgroups across data types and sources. We then examined the molecular and clinical features of these two groups. One subgroup (BPH-A) was enriched in stromal signatures[36] (Fig. 3b and Supplementary Fig. 9), again in the validation cohort as well (Fig. 3d and Supplementary Fig. 10), consistent with the presence of stromal signatures in previous reports[18]. Of note, there was no clear enrichment of stromal cell content visible on histopathology analysis of these samples, suggesting that molecular characterization provided

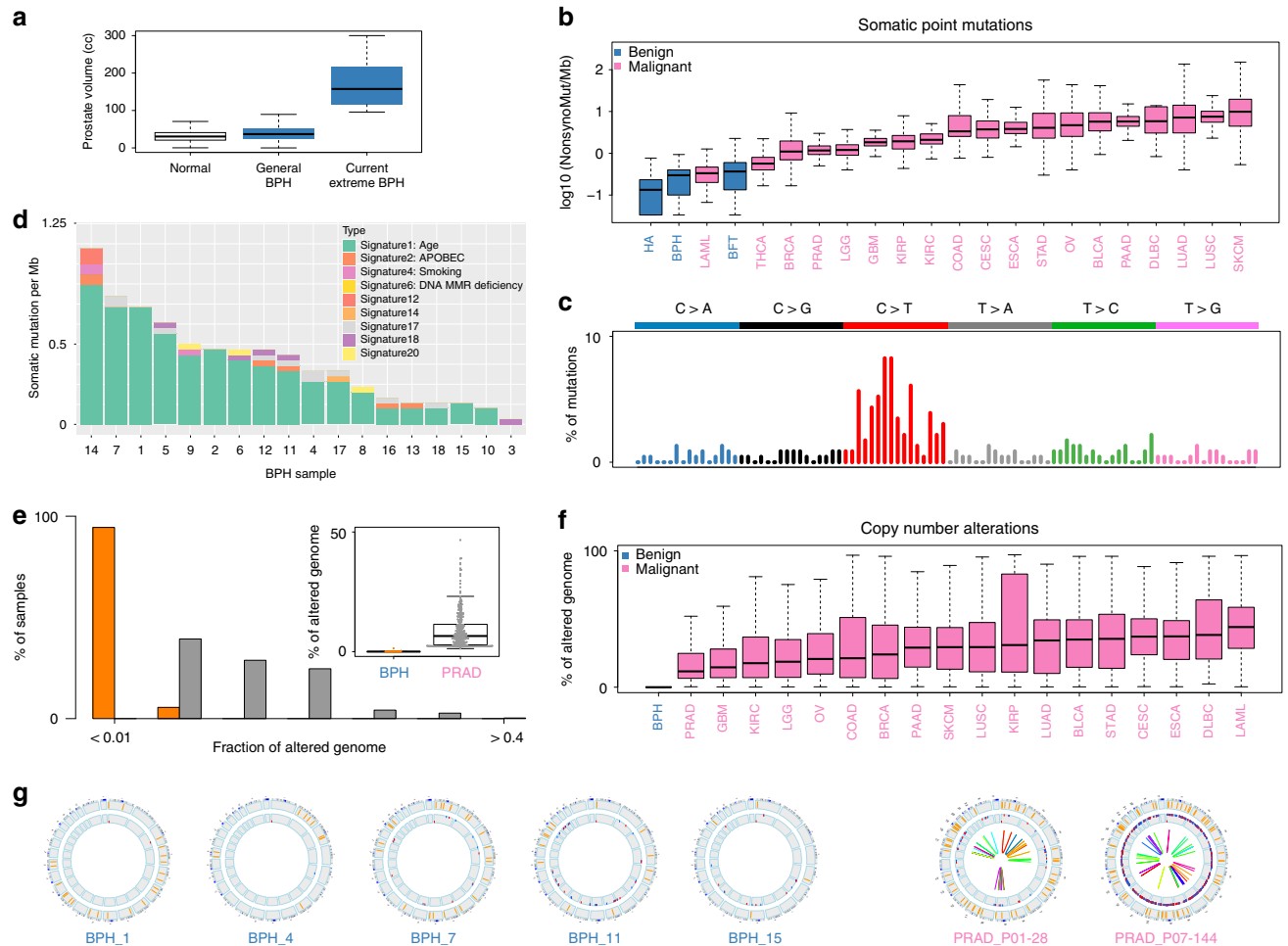

**Fig. 1 Minimal genomic alterations are found in BPH samples. a** Boxplots of prostate volume (cc) of normal ($n = 1427$), general BPH ($n = 2950$), and extreme BPH ($n = 18$) cases used in the current study. **b** The prevalence of somatic non-synonymous mutations across benign disease and multiple cancer types. The y-axis represents the log10 value of mutations. The x-axis includes benign (blue) and malignant tumors (pink) from TCGA studies. BLCA ($n = 130$); BRCA ($n = 988$); CESC ($n = 40$); COAD ($n = 216$); DLBC ($n = 48$); ESCA ($n = 184$); GBM ($n = 282$); KIRC ($n = 213$); KIRP ($n = 113$); LAML ($n = 196$); LGG ($n = 533$); LUAD ($n = 543$); LUSC ($n = 178$); OV ($n = 243$); PAAD ($n = 185$); PRAD ($n = 499$); SKCM ($n = 472$); STAD ($n = 289$); THCA ($n = 504$). HA: hepatocellular adenomas ($n = 46$), and BFT: breast fibroadenomas ($n = 30$). **c** The somatic mutation signatures of BPH. The signature is based on the 96 substitutions classification defined by the substitution class and sequence context immediately 3' and 5' to the mutation position. The y-axis represents the percentage of mutations attributed to a specific mutation type. The six types of substitutions are shown in different colors. **d** The contribution of mutation signatures to each BPH sample. Each bar represents a BPH case and y-axis denotes the number of somatic mutations per megabase. **e** The fraction of altered genome, partitioned into bins covering a range from <0.01 to ≥0.4, shown as a histogram for BPH and primary prostate cancer samples. Inset: boxplot of altered genome fraction for BPH samples ($n = 18$) and primary prostate cancer ($n = 333$) samples from TCGA study. **f** The lower fraction of altered genome in BPH (blue) when compared to malignant diseases (pink) from TCGA studies. BLCA ($n = 130$); BRCA ($n = 988$); CESC ($n = 40$); COAD ($n = 216$); DLBC ($n = 48$); ESCA ($n = 184$); GBM ($n = 282$); KIRC ($n = 213$); KIRP ($n = 113$); LAML ($n = 196$); LGG ($n = 533$); LUAD ($n = 543$); LUSC ($n = 178$); OV ($n = 243$); PAAD ($n = 185$); PRAD ($n = 499$); SKCM ($n = 472$); STAD ($n = 289$); THCA ($n = 504$). **g** Circos plots of 5 BPH and 2 primary prostate cancer samples. The rings from outer to inner represent somatic coding mutations, copy number alterations and genomic rearrangements respectively. Definition of box plots in panels 1a, b, e and f: the center line represents median value, box limits represent 25% and 75% quantiles, and the top and bottom lines represent minimal and maximal values, respectively.

independent information (Supplementary Fig. 11). To gain further insight into the potential cellular components of these molecular data, we examined the signatures of different cellular compartments from recently reported single cell RNA-seq from normal prostate tissue (Supplementary Fig. 12)[37]. This data further confirmed the overall stromal enrichment in BPH-A subgroup, but also highlighted distinct compartments with clear similarities and differences between subgroups, reinforcing the concept that both distinct cellular components and gene expression changes within compartments may contribute to biologically informative molecular signals.

The second subgroup (BPH-B) was enriched for patients with obesity (BMI > 30) and hypertension (Fig. 3d), potentially suggesting distinct pathobiology. Consistent with this, gene set enrichment analysis between the two subgroups demonstrated significant differences among metabolism related signatures, such as fatty acid and amino acid metabolism (Fig. 3d and Supplementary Fig. 13). Positive correlation of metabolism dysregulation between the two subgroups extended to both cohorts (Fig. 3d and Supplementary Fig. 13), consistent with the clinical associations with obesity and hypertension. We then explored signaling pathways within each subgroup to further

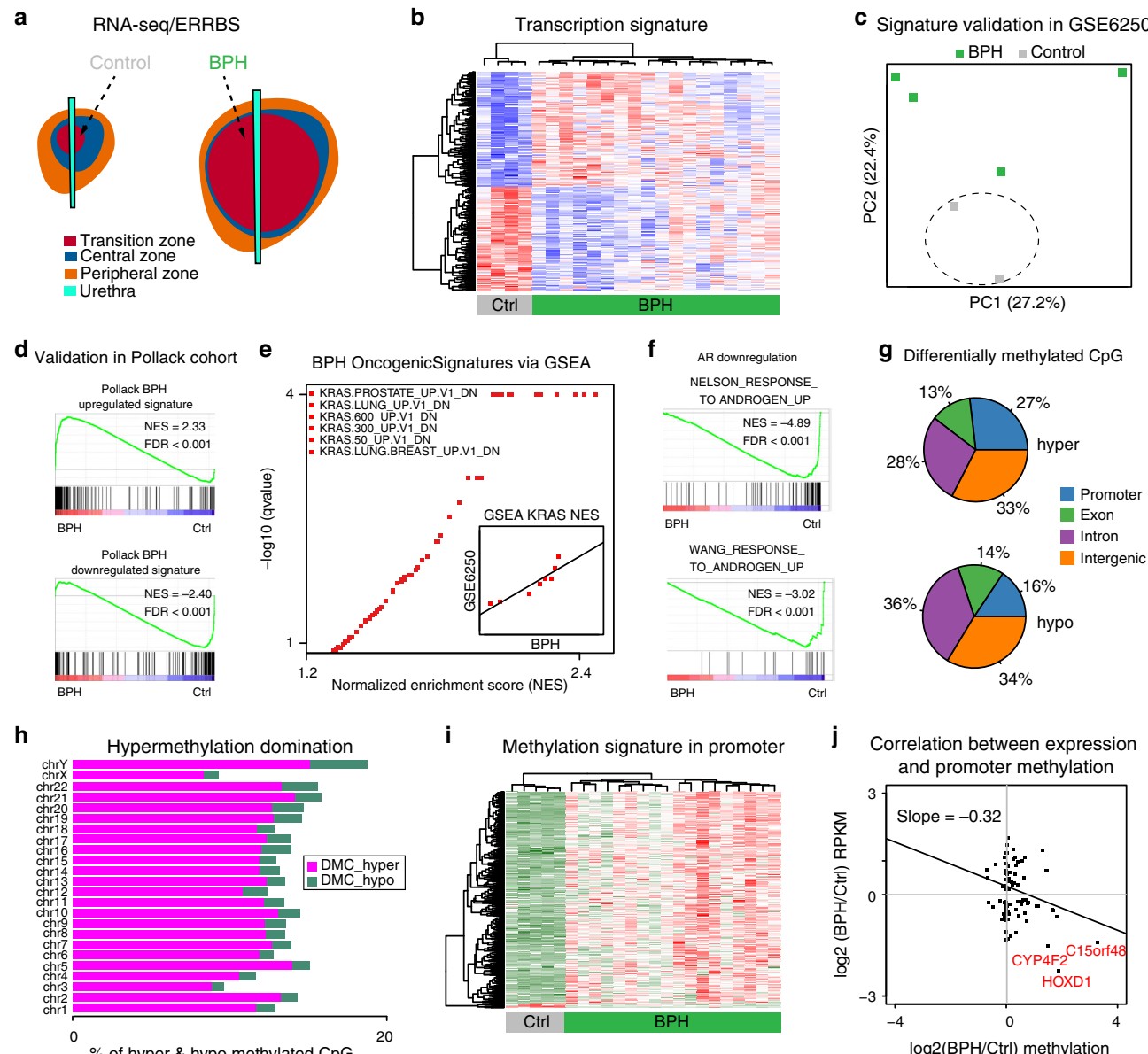

**Fig. 2 BPH transcription and methylation profiles. a** Diagram of sampling location of BPH and control samples used for RNA-seq and ERRBS. Green color represents BPH samples, and gray color represents control samples from normal transition zones of men undergoing radical prostatectomy for prostate cancer (without clear evidence of BPH). **b** Hierarchical clustering and heatmap of transcriptional signatures based on significantly differentially expressed genes between BPH and control samples. **c** Validation of transcriptional signature (panel B) in an independent study (GSE6250). Green color represents BPH samples, and gray color represents control samples from normal transition zones of men undergoing radical prostatectomy for prostate cancer (without clear evidence of BPH). **d** The concordance of transcriptional signatures between current and previous BPH study. GSEA analyses of current BPH cases showing that genes upregulated in previous BPH cases are positively enriched, and genes downregulated in previous BPH cases are negatively enriched. **e** GSEA analysis of BPH cases in oncogenic signatures showing that genes downregulated in many cell lines when over-expressing an oncogenic form of *KRAS* gene are positively enriched. Similar results and high correlation with GSE6250 study are shown in inner panel. **f** GSEA analysis of BPH cases in AR related signatures showing that genes upregulated in LNCaP cells treated with synthetic androgen are negatively enriched. **g** Pie chart of differentially methylated CpGs between BPH and control samples among different genomic regions. Colors denote different genomic related regions. **h** Hypermethylation domination found in each chromosome. The x-axis represents the percentages of hypermethylated (label as pink) and hypomethylated (label as green) CpGs. **i** Hierarchical clustering and heatmap of promoter methylation signature between BPH and control samples. **j** Correlation between transcription and promoter methylation signatures, examples of epigenetically silenced genes shown in red.

understand the underlying biology. As compared to control samples, we found differential expression of metabolism related genes predominantly in BPH-A samples (Fig. 3f, g, Supplementary Tables 13 and 14), consistent with the metabolism difference between two subgroups. Unbiased GSEA showed multiple deregulated pathways for each subgroup, with many pathways were negatively correlated between two subgroups (Fig. 3h and Supplementary Table 15), reinforcing distinct biology. Together these molecular data suggested two distinct biological categories of BPH—one with stromal-like molecular features, and the other associated with deregulation of metabolic pathways that presents in patients with underlying metabolic disturbances.

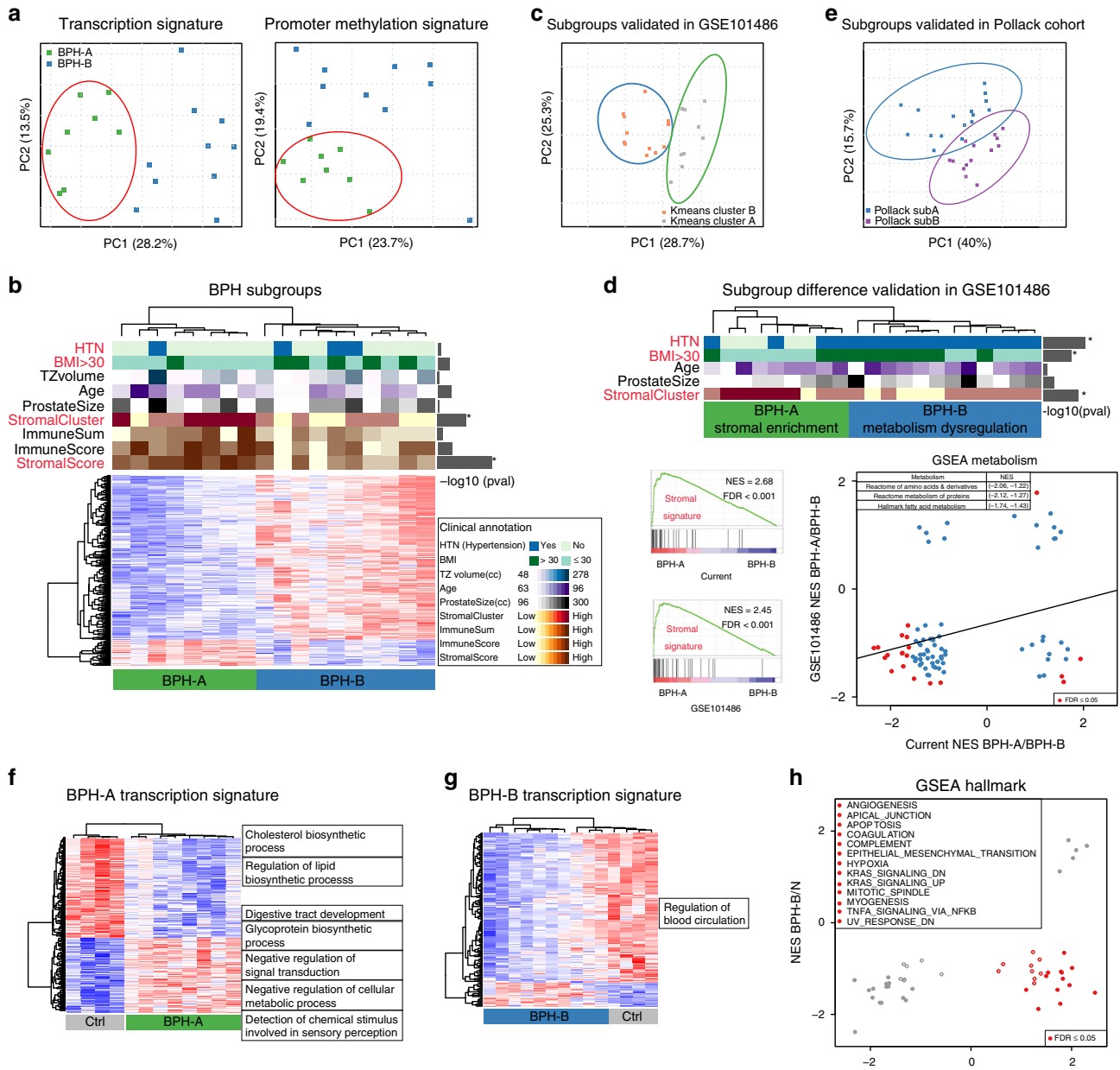

**Fig. 3 Identification and validation of distinct BPH subgroups. a** Principal component analysis (PCA) based on transcriptional and promoter methylation signatures on RNA-seq data. Green denotes subgroup A, and blue denotes subgroup B. **b** The clinical and biological differences between two BPH subgroups: BPH-A ($n = 8$) and BPH-B ($n = 10$) via using two-sided Fisher's exact test and Wilcoxon signed-rank test. *p-value < 0.05 assessing differences between two BPH subgroups. HTN (hypertension): p-value = 0.588; BMI > 30: p-value = 0.151; TZ (transition zone) volume: p-value = 0.728; Age: p-value = 0.119; Prostate size: p-value = 0.789; Stromal cluster: p-value = 0.011; Immune genes zcores: p-value = 0.633; Immune score: p-value = 0.101; Stromal score: p-value = 0.002. **c** The validation of BPH subgroups on an independent microarray study GSE101486 with 21 BPH samples via principal component analysis (PCA). K means clustering identified two distinct subgroups based on BPH subgroup signature from panel b. **d** Clinical and biological differences are shown between two subgroups: BPH-A ($n = 9$) and BPH-B ($n = 12$) from GSE101486 study via using two-sided Fisher's exact test and Wilcoxon signed-rank test. *p-value < 0.05 assessing differences between two BPH subgroups. HTN: p-value = 0.045; BMI > 30: p-value = 0.035; Age: p-value = 1; Prostate size: p-value = 0.395; Stromal cluster: p-value = 0.014. Bottom left represents GSEA plots of significant enrichment of stromal signature in subgroup BPH-A when comparing with subgroup BPH-B from both current and GSE101486 studies. Bottom right showed the correlation of metabolism dysregulation between two subgroups. The x-axis denotes the normalized enrichment scores from current study, and y-axis denotes the normalized enrichment scores from GSE101486 study. Red dots represent the significant signature with FDR < 0.05 in either one of two studies. Examples of metabolism dysregulation are shown. *p-value < 0.05 assessing differences between two BPH subgroups. **e** The validation of BPH subgroups on an independent study[18] with 30 BPH samples via principal component analysis (PCA), based on BPH subgroup signature from panel b. **f** Hierarchical clustering and heatmap of transcriptional signature between subgroup BPH-A and control samples. **g** Hierarchical clustering and heatmap of transcriptional signature between subgroup BPH-B and control samples. **h** The difference of enriched pathways between BPH subgroups when comparing with control samples. Red dots indicate the difference of MSigDB hallmark signatures via GSEA with FDR ≤ 0.05 between two BPH subgroups. The x and y-axes represent the normalized enrichment score of signatures from each BPH subgroup when comparing to control samples.

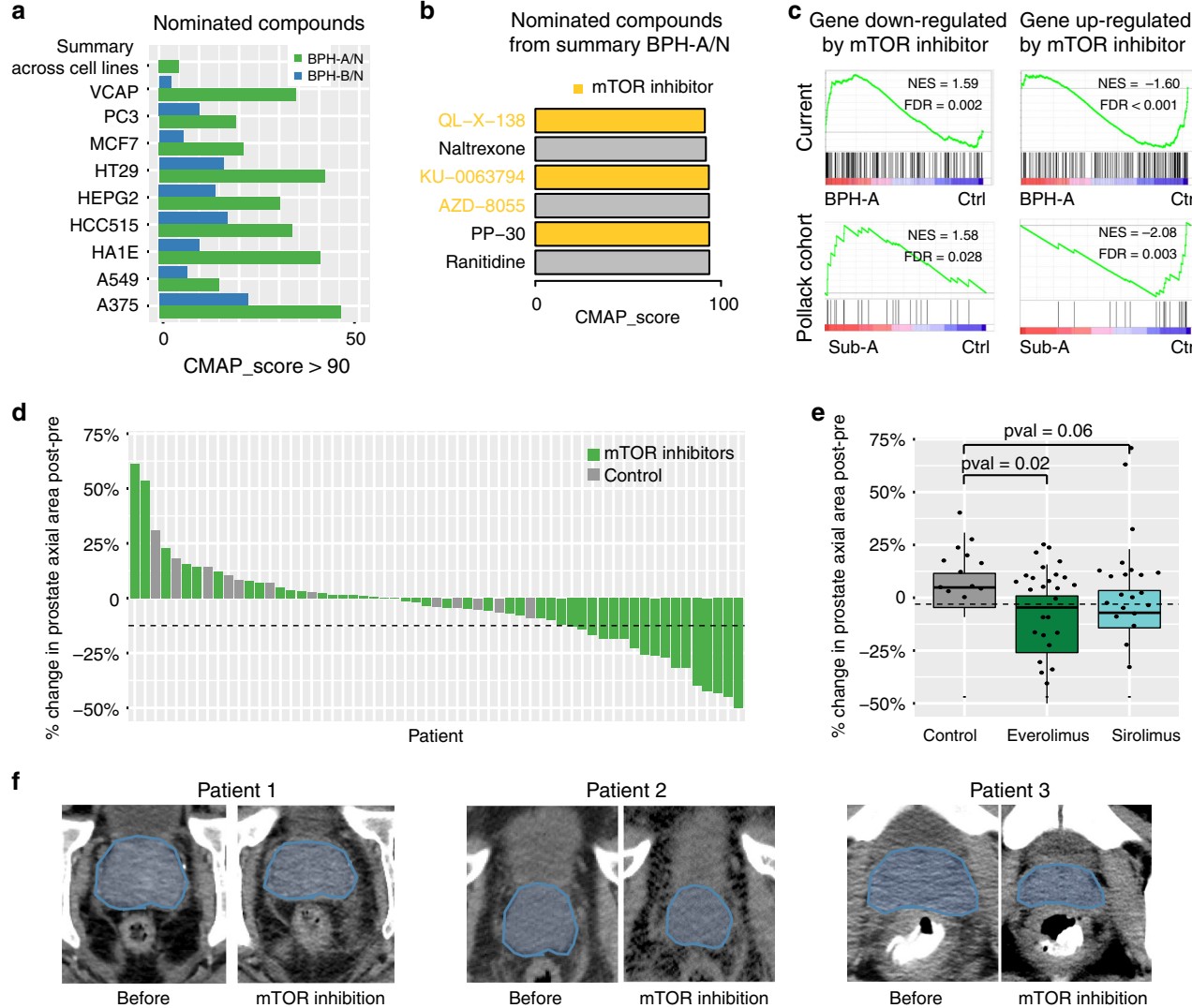

**Fig. 4 BPH subgroup-specific alterations and potential therapeutic options. a** Barplots of nominated compounds from each BPH subgroup when comparing with control samples across multiple cell lines, and summary from all cell lines via Connectivity Map (CMAP). The *x*-axis denotes CMAP score. Different colors represent BPH subgroups. **b** Nominated compounds from subgroup BPH-A via Connectivity Map. The *x*-axis denotes the CMAP score. **c** GSEA analysis of mTOR related signatures in subgroup BPH-A and subgroup Sub-A from independent study[18], showing that genes down-regulated by mTOR inhibitor are positively enriched, and genes up-regulated in CEM-C1 cells (T-CLL) by mTOR inhibitor are negatively enriched, compared to control samples. **d** Waterfall plot of % prostate axial area change on computed tomography in 47 patients after initiating therapy with an mTOR inhibitor and 12 kidney stone patients (negative controls). Different colors represent patient type. Dashed line represents a predetermined threshold (12.5%) for a significant decrease in area. **e** Boxplots of % prostate area change in 47 patients taking mTOR inhibitors (26 with Everolimus and 21 with Sirolimus treatments), and 12 negative controls. *P*-values represent prostate size change for each drug as compared to controls via using two-sided Wilcoxon signed-rank test. The center line represents median value, box limits represent 25% and 75% quantiles, and the top and bottom lines represent minimal and maximal values, respectively. **f** Examples of CT scans from three patients who had a decrease in prostate size after initiation of an mTOR inhibitor. Prostate highlighted in blue.

**mTOR inhibition as subtype specific therapy**. To nominate potential subtype specific therapeutic options, we utilized the Connectivity Map[38,39] analysis (Fig. 4a and Supplementary Table 16), which uses transcriptional expression data to probe relationships between diseases, cell physiology, and therapeutics. Strikingly, we found 50% of nominated compounds in BPH-A subgroup were related to inhibition of mTOR signaling (Fig. 4b), and the subgroup enrichment of mTOR signaling was validated in two independent cohorts (Fig. 4c), consistent with prior isolated reports in model systems[40,41]. To interrogate the potential effect of mTOR treatment on the prostate, we examined prostate size on cross-sectional imaging in patients taking mTOR inhibitors. We identified 425 male patients who had been prescribed an mTOR inhibitor (everolimus, sirolimus, or temsirolimus) for transplant or treatment of a non-prostate malignancy. We then reviewed these patient's charts to identify men with accessible CT imaging including the pelvis before and after treatment, identifying 47 such subjects. CT scans from these 47 subjects and 12 men with serial imaging for nephrolithiasis (negative controls) then underwent blinded review and assessment of prostate size (Supplementary Fig. 14). Of these men, 17/47 had a prostate size decrease based on pre-established thresholds (12.5% decrease from baseline), all of whom were on an mTOR inhibitor (Fig. 4d, f). None of the nephrolithiasis patients

showed a significant decrease in prostate size. A higher proportion of patients taking Everolimus had a decrease in prostate size ($p$-value = 0.02), compared to kidney stone patients (0%) (Fig. 4e). Similar trends were seen in the effect of mTOR inhibitors on absolute cross-sectional area (Supplementary Fig. 15). Overall, these data support that molecularly identified BPH subgroups represent biologically distinct subtypes of disease. Our analysis suggests one subgroup as dependent on mTOR signaling, and we show here in exploratory clinical analysis that mTOR inhibition affects prostate size, thereby nominating a potential novel therapeutic option.

## Discussion

In summary, we report a comprehensive, multi-level molecular investigation of BPH, including genomic, transcriptomic and epigenomic profiling. While dogma often suggests BPH represents a benign neoplastic process, we find no evidence of somatic genomic alterations, unlike benign neoplasms like such as frequent *MED12* mutations in breast fibroadenomas[20,21] and uterine fibromas[24] or *FRK* mutations in hepatocellular adenomas[22], and BPH exhibited an age-related mutation signature, consistent with the higher prevalence in older patients as opposed to underlying oncogenic processes. While we can detect no somatic changes consistent with driver genomic alterations, given technological and sampling limitations, we cannot completely rule out the presence of genomic alterations below limits of detection. Furthermore, unlike the global hypomethylation signature in neoplastic diseases[42–44], the DNA methylation landscape in BPH was dominated by hypermethylation. Together, our genomic and epigenomic data argues against BPH arising from a neoplastic disease process. However, BPH is a complex, heterogeneous, and pleomorphic disease that can be defined in many ways, likely representing a collection of distinct etiologies and pathobiologies[3,45,46]. Here, the BPH cases we interrogated were of extreme size (top 1 percentile of prostate size), which we hypothesized would be most likely to harbor informative molecular events, but may not be generalizable to more routine cohorts.

By integrating the transcriptional and DNA methylation data, we identified and validated two molecular subgroups in BPH, one characterized by a stromal signal (despite no clear differences in histology), and the other associated with hypertension and obesity, which was consistent with metabolism dysregulation between these two subgroups. Moreover, the altered signaling pathways of each subgroup comparing with control samples were related to the metabolism regulation and hypertension.

Molecular analysis suggested distinct patterns of therapeutic vulnerability for each BPH subgroup. Specifically, this nominated mTOR inhibitors as having preferentially activity in one subgroup. Using retrospective clinical data, we found 17/47 patients treated with mTOR inhibitors showed significant decrease in prostate size. Despite this promising signal in clinical data, the conclusion that mTOR inhibitors can affect prostate size should be tempered by the limitations of this analysis[47,48]. First, these are retrospective data with no ability to examine the molecular features of the responding patients. Future rigorously designed clinical studies, with the ability to stratify patients by molecular subgroup, will be necessary to validate and move these findings further toward clinical deployment. Second, additional mechanistic validation is will be needed to clarify the nature of mTOR involvement and the specific signaling pathways in play. mTOR is a critical signaling pathway for nutrient sensing and cell growth, and has complex cross talk with other signaling pathways, including androgen receptor signaling.

Overall, these data support that molecularly identified BPH subgroups represent biologically distinct subtypes of disease. Our analysis suggests one subgroup is dependent on mTOR signaling, and we show here in exploratory clinical analysis that mTOR inhibition affects prostate size, thereby nominating a potential novel therapeutic option.

## Methods

**Samples collection.** Patients with BPH were prospectively enrolled for sequencing of prostate tissue samples from transition zones under a protocol approved by the institutional review board of Weill Cornell Medical College. Slides were cut from frozen blocks, stained with hematoxylin and eosin, and areas for coring were designated. Slides were annotated for epithelial and stromal content by expert GU pathologists. This was a single center BPH study from Weill Cornell Medicine, and these were not consecutive cases. Patients with prostate size meeting inclusions criteria underwent informed consent for this IRB approved protocol prior to surgical therapy for BPH. All the BPH samples were derived from surgical specimens and prospectively collected and banked as frozen tissue. Normal controls for RNA-seq and ERRBS were obtained from men undergoing radical prostatectomy for prostate cancer without BPH. Under the supervision of study pathologists, benign areas of transition zone distant from the tumor were cored, and molecular analysis confirmed the absence of any cancer related molecular alterations (Supplementary Table 8 and Supplementary Figs. 3 and 4). Written informed consent was obtained, including discussion of risks associated with germline sequencing. Fresh tissue samples were collected and processed using internal standard operating procedures.

**DNA sequencing, data processing and analysis pipeline.** Whole genome sequencing (WGS) on 5 BPH samples with matched normal samples from blood tissue was performed in New York Genome Center under standard protocol and pipeline for 100 bp paired-end sequencing. Samples were sequenced with average genome coverage of 100× for BPH samples, and 50× for matched control samples. Whole exome sequencing (WES) on 13 BPH samples with matched control samples from blood tissue was performed in the Genomics Core of Weill Cornell Medicine under standard protocol and pipeline for 75 bp paired-end sequencing. Whole exome sequencing capture libraries were constructed from BPH and control tissue by using SureSelected Exome bait (Agilent), and samples were sequenced with average target exon coverage of 300-360x for BPH and matched control samples.

Paired-end sequence reads of WGS and WES data were aligned to the human reference genome (hg19) using BWA[49] v0.7.12. Sorted bam files were generate via SAMtools[49] v.0.1.19, and the duplicated mapped reads were marked with Picard v1.134. BAM files were locally realigned to the human reference genome using GATK[50] v3.7, and somatic base substitutions and small indels were detected by using MuTect[51] v1.1.4 and Varscan2[52] v2.3.9, with the sequencing coverage cutof of at least 14x in BPH and 8x in control samples. Mutations were defined as the shared output between MuTect and Varscan2. After excluding the known human SNPs (dbSNP Build 150) and SNPs detected from control samples, the remaining mutations were annotated by ANNOVAR[53] v2018.04.16 with GENCODE (v19) human gene annotation. The mutation signature was detected by using SomaticSignatures[54] v2.20.0. DELLY[55] v0.8.1, BreakDancer[56] v1.3.6 and CREST[57] v2016.12.07 were used to identify the genomic translocations. The mutations results of malignant diseases were downloaded from TCGA studies (https://portal.gdc.cancer.gov/) via gdc-client tool. The mutation results of hepatocellular adenomas (HA) and breast fibroadenomas (BFT) were derived from published studies[20–22].

**Copy number analysis.** DNA from BPH and matched control samples from blood tissue were analyzed by Affymetrix SNP 6.0 arrays to detect the regions of somatic copy number alteration. Copy number estimates were performed via using Circular Binary Segmentations[58], and significant copy number alterations were identified from segmented data using GISTIC (v2.05)[59]. The copy number alterations of malignant diseases were downloaded from TCGA portal (https://portal.gdc.cancer.gov/) via gdc-client tool. Segments with log$_2$-ratio >0.3 were defined as genomic amplifications, and log$_2$-ratio < −0.3 were defined as genomic deletions.

**RNA-seq processing and analysis pipeline.** RNA-seq library for BPH and control samples from patients without BPH symptoms were generated using Poly-A and Ribo-Zero kits. RNA-seq was performed in the Genomics Core from Weill Cornell Medicine under standard protocol and pipeline for 75 bp paired-end sequencing. Reads were mapped to the human reference genome sequence (hg19) using STAR[60] v2.4.0j. Then the resulting BAM files were subsequently converted into mapped-read format (MRF) using RSEQtools[61] v0.6. The read count of each gene was calculated via HTSeq[62] v0.11.1 using GENCODE (v19) as reference gene annotation set. Quantification of gene expression was performed via RSEQtools[61] v0.6, and expression levels (RPKM) were estimated by counting all nucleotides mapped to each gene and were normalized by the total number of mapped nucleotides (per million) and the gene length (per kb). Fusion genes were detected via FusionSeq[63] v0.1.2. Combat[64] v3.20.0 was used to remove the batch effect of different RNA-seq libraries for the downstream gene expression analysis. Heatmap

and hierarchical clustering were performed via using correlation distance and Ward's method. GSEA[31] v3.0 was performed using JAVA program and run in pre-ranked mode to identify enriched signatures. The GSEA plot, normalized enrichment score and q-values were derived from GSEA output for hallmark signature, and the metabolism related signatures were derived from MSigDB[65] v6.2 database. Differential expression analysis was performed using the Wilcoxon signed-rank and F-statistic test after transforming the RPKMs via log2(RPKM + 1). Multiple-hypothesis testing was considered by using Benjamini-Hochberg (BH; FDR) correction. The immune score and stromal score were calculated from gene expression of BPH samples via ESTIMATE[66] v2.0.0. The ImmuneSum was defined as the sum of normalized z-scores from gene expression of immune markers including *PDCD1*, *PDCD1LG2*, *CD274*, *CD8A* and *CD8B*. The comparison of molecular and clinical features between two subgroups was performed using Fisher's exact test and Wilcoxon signed-rank test. The compounds for each subgroup were identified by using Connectivity Map (CMAP)[39] with the top most overexpressed and underexpressed genes as the input, and CMAP score >90 were used to select the nominated compounds.

**DNA methylation processing and analysis pipeline**. Genomic DNA was isolated from BPH and control samples, and submitted to the Epigenomics Core of Weill Cornell Medicine under standard protocol and pipeline. The Epigenomics Core facility in Weill Cornell Medicine supported alignment and methylation extraction for ERRBS data[67]. Differentially methylated CpGs (DMCs) were identified by methylKit[68] and RRBSeeqer[69] (false discovery rate = 5%, and methylation difference more than 10%). Differentially methylated regions (DMRs) were defined as regions containing at least five DMCs within 250 bp window. Genomic regions for CpGs were defined according to the following definitions. CGIs (CpG islands) were defined using annotations from RefSeq. CGI shores were defined as the regions encompassing 1 kb upstream and downstream of known CGIs. Non-CGIs were defined as regions at least 10 kb away from known CGIs. Promoters were defined as the regions encompassing 2 kb upstream and downstream of the TSS (transcription start site) of RefSeq genes. Promoter methylation for each gene was calculated by averaging the methylation levels of all CpGs covered in the promoter.

**Effect of mTOR inhibition on prostate size**. We searched our electronic medical record to identify all adult male patients who received therapy with an mTOR inhibitor (Sirolimus, Everolimus, or Temsirolimus) using our institutional i2b2 search tool (IRB 1510016681R003) (Supplementary Fig. 9). These patients were not selected based on a preexisting history of any specific disease state, symptoms, or age. The median age of patients at the time of initial CT scan was 58 years of age, and at follow up CT scan was 60 years of age. Records were manually reviewed in order to identify individuals with CT imaging containing the pelvis before and after therapy. The most proximate CT scan prior to the initiation of therapy and the CT scan as close to 6 months after the initiation of therapy were used. This interval was chosen based on the known time course of prostate size changes in response to finasteride[70]. As negative controls, we identified 12 kidney stone patients over age 35 at the time of baseline CT who had serial CT imagining including the pelvis who did not take 5-alpha-reductase inhibitors, have prostate cancer, recurrent urinary tract infections, or a history of prostatic surgery. In order to establish a signal window, we performed an initial unblinded pilot including patients who underwent androgen deprivation therapy as a positive control and kidney stone patients as a negative control. We determined that a decrease in prostate size of >12.5% in sequential CT scans would have captured 9/10 androgen deprivation therapy patients from CT scans spaced ~6 months apart, and would be >2 standard deviations from the mean decrease in prostate size of the kidney stone patients.

We then extracted accession numbers from both kidney stone and mTOR inhibitor patients and using a random number generator placed them in arbitrary order for review. A radiologist then reviewed these scans by using accession numbers unaware of treatment assignment (mTOR inhibitor or kidney stone) or whether it was a baseline or follow up study. The prostate was measured in two dimensions in the axial slice with the greatest apparent prostate area, with area computed as anterior–posterior × transverse measurements. When unclear, the prostate was measured in two axial slices and the maximum area utilized.

Following review, scans were then re-identified, and subjects with a baseline axial prostate size <1000 mm² were excluded from further analysis. Subjects where the blinded review showed a >12.5% decrease in area, defined as initial area- follow up area)/ initial area, then underwent a subsequent blinded review by a urologist (JS). Agreement was necessary between both reviews for a decrease to be considered true: when urology review did not identify a decrease >12.5% and this differed from radiology review by <20% urology review was prioritized. For subjects where both reviewers agreed on the decrease in area, initial radiology review dimensions were utilized. When there was a >20% discrepancy in measurements (irrespective of degree), scans were re-reviewed blinded by radiology, and these measurements utilized.

**Reporting summary**. Further information on research design is available in the Nature Research Reporting Summary linked to this article.

## Data availability

The SNP array data has been deposited in GEO under the accession GSE124187 (https://www.ncbi.nlm.nih.gov/geo/query/acc.cgi?acc=GSE124187), RNA-seq data has been deposited in GEO under the accession GSE132714 (https://www.ncbi.nlm.nih.gov/geo/query/acc.cgi?acc=GSE132714), and ERRBS data has been deposited in GEO under the accession GSE123111 (https://www.ncbi.nlm.nih.gov/geo/query/acc.cgi?acc=GSE123111).

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

## Acknowledgements

We are grateful to the benign prostatic hyperplasia patients and families who contributed to this research. We thank the WCM Genomics and Epigenomics Core Facility, and all the surgeons, pathologists, research coordinators, and trainees to contributed to patient enrollment and tissue collection. This work was supported by: NCATS (CTCS: UL1 RR 024996), a Urology Care Foundation Rising Star in Urology Research Award (C.E.B.), Damon Runyon Cancer Research Foundation MetLife Foundation Family Clinical Investigator Award (C.E.B.), the Prostate Cancer Foundation (C.E.B), the Prostate Cancer Foundation Young Investigator Award (D.L), the Frederick J. and Theresa Dow Wallace Fund of the New York Community Trust (J.S.), and Damon Runyon Cancer Research Foundation Physician Scientist Training Award (J.S.).

## Author contributions

C.E.B., A.S. and D.L. designed research studies. D.L., J.S., R.S.G., D.R., A.V., D.P., V.R., J.F., H.P., D.L., D.T., K.S., Z.W., A.T., R.L., B.C., A.F.O. and J.M.M. conducted experiments and acquired data. D.L., J.S., C.E.B. and A.S. analyzed the data. D.L., A.S. and C.E.B. wrote the manuscript. J.S., A.R., A.T., R.L., B.C., A.F.O., J.M.M., F.D., O.E. and M.A.R., helped to revise the manuscript.

## Competing interests

The authors declare no competing interests.
