## [Peer Review File · Nature Communications]

Reviewers' Comments:

Reviewer #1:

Remarks to the Author:

In this study, Liu et al. sought the underlying molecular features of the transition to BPH via multi-modality molecular characterization of 18 cases. This is a well-performed study addressing an important question. The profiling is not particularly innovative, but comprehensive and the conclusions reasonable and well-written. While the molecular data prove largely negative (which is perhaps interesting in and of itself), the transcriptional information was information-rich and validate in part previously known facets of BPH. This is laudable attempt to answer an outstanding question in the urology/prostate cancer field. Findings are likely limited with respect to broader conclusions regarding benign precursor lesions in other settings, etc. Major comments are:

1.) Regarding the cellularity of the specimens, can the authors highlight in greater detail their confidence in not having sequenced largely normal cells? Cellularity estimates are often discordant between pathology and analytical estimates from data, and this reviewer recognizes the complexity of doing so for non-tumor specimens in this case that largely lack the CNAs or molecular changes on which these estimates are typically based, but somatic mutations in largely diploid specimens (even if not drivers), may assist here? Such an analysis would be especially helpful in showing that the two RNA subgroups are different for reasons other than cellularity.

2. If the normal controls were from RPs of men with PCa but not BPH, were these samples investigated alone and in detail in the molecular data to confirm they were normal prostate (no evidence molecular changes consistent with prostate cancers)? Such data would be helpful, its related to the interpretation of data throughout the manuscript, and would generally increase the confidence in the analysis.

3. Ostensibly, given the prevalence of BPH compared to primary prostate cancer (and the spectrum and frequency of drivers in the latter), it is unlikely to manifest clonal drivers per se, but what about evidence of subclonality in the data? Can the authors reasonably preclude the possibility of potential rare subclones (within reason vis a vis sensitivity) of neoplastic disease (perhaps in the cases with elevated PSA, or similar)? This is unlikely possible from the sequencing they have performed to date (due to the limited sensitivity of lower coverage broad sequencing), but perhaps they can examine the data for evidence of subclones at reasonably detectable frequencies?

4. Is there absolutely no evidence of ERG fusion, even rarely, in the RNA? Would be a nice negative control to just show. Same goes for all known PCa drivers, of which the number is limited. A fill-out of sequencing reads at these loci in WGS/WES/RNA to show absolutely negative even at sub-detection thresholds would be nice to show.

5. Some images of the histology of cases (by stromal signal and not) would be appreciated, perhaps as supplementary materials to reaffirm the statement about the signature itself.

6. The clinical data regarding mTOR inhibition is intriguing and a creative use of retrospective data, but as such, confounded by multiple factors that are difficult to control for in this kind of analysis (as the authors are well-aware). Moreover, the statistics here are exploratory and not consistent with proper univariate and multivariable outcome analyses. For those reasons, care should be taken in how strongly to articulate the conclusions of this analysis and the viability of mTOR inhibition as a therapeutic strategy for BPH without more data.

7. In the absence of some degree of mechanistic confirmation (perhaps in organoids from BPH patients?) it's difficult to get a sense for validity of the mTOR association and some of the pathways suggested. This reviewer recognizes this is likely outside the scope of the present study and possible future work for the team, but its absence does limit the rigor and robustness of the

conclusions and should likely be acknowledged in a conservative discussion of their findings.

Reviewer #2:

Remarks to the Author:

In this well-written manuscript the authors describe the molecular characterization of benign prostatic hyperplasia (BPH) using a variety of sequencing based approaches, including whole genome sequencing, transcriptome sequencing and methylation analysis. The cohort consists of 18 extreme cases of BPH as assessed by volume. Control samples from transitional zone and peripheral zone of normal prostate were used in some of the analyses. The study has generated a rather large dataset from BPH specimens and is the first whole genome sequencing description of this disease. While many of the findings in this regard are negative, (no structural rearrangements, low mutation burden lacking candidate drivers), this information is still important to our understanding of the biology of BPH. Results from methylation analyses and, to some degree, transcriptome analyses recapitulate previous studies. The inclusion of clinical data regarding prostate volume reduction following MTOR inhibition is intriguing, but is not linked directly to the molecular data generated by this study.

The manuscript is concise and does integrate multiple data types well. This reviewer has some concerns to address and further clarifications to request.

Overall the data and the analytical results are presented very well. The conclusions follow the data with a few exceptions mentioned next. These are the principal concerns.

The categorization of the Subgroup B (BPH-B) is not clearly presented in the text or Figures. While it is different than Subgroup A in terms of the expression of genes coding for elements of metabolic pathways, it is not shown clearly to differ from controls. In fact, Subgroup A is shown to be different than controls for expression in metabolic pathways and then many of the following figures and supplement follow the theme that A is different than B, which, while true, does not support that B is different than controls. One could alternatively conclude that BPH-A differs from normal in both stromal signature and expression of metabolic pathways. An analysis and heat map representation including BPH-A, BPH-B and controls would help to clarify gene sets which might be useful to distinguish the 3 proposed classes.

The link between Subgroup A molecular profile and response to MTOR inhibition is not firmly established. This reviewer could not find any transcriptional characterizations of the responders or non-responders to MTOR inhibition. This can be understood, of course, as the MTOR clinical data is retrospective, without characterization of putative transcriptional classes. This reviewer does not know if archival biopsy material remains from those MTOR cases, but sequencing and analyses of such material would provide support for one of the major claims of the paper. Specifically, "found that MTOR inhibitors may be preferentially active in one subgroup" is not supported yet by the data in the paper - yet these conclusions are suggested in the title. The MTOR pathway has been known to regulate both organ and cell size in normal tissues (PMID:12080086 and 22575479) and the responding cases in this paper have not been shown to have aberrant MTOR signaling.

A lesser concern is the discrepancy between the stromal transcriptional signature and the lack of histological correlation. The authors do note this, but could some investigations be done to address this apparent discrepancy? Could the authors perform some IHC experiments to address the question could the stromal signature be due to a more "active" stroma, i.e. is the expression of some of the signature genes in the stromal panel higher at the individual cell level in BPH vs normal?

This reviewer commends the authors on their inclusion and analyses of relevant external datasets. Two lesser points- could the authors be more explicit that a stromal signature was previously found in BPH in the Pollack study. Second, could an attempt be made to analyze the proposed subgroup B vs the single cell data of Henry et al?

Minor points.

1. The age of men in the description of the frequency of BPH is missing (line 83)
2. Principal component analysis is incorrectly called Principle component analysis at several places in the text and Figure 3.
3. A more explicit depiction of variant allele frequencies and perhaps clonality assessment would be informative to understanding the biology of BPH, could the authors add this?
4. The Figures are very nicely prepared-, there is a small problem with printing Figure 2, some of the captions and labels in the Figure print with a nonsensical font.

Reviewer #3:

Remarks to the Author:

The manuscript "Integrative genomic, transcriptomic, and epigenomic analyses of benign prostatic hyperplasia reveal new options for therapy" by Liu et al., is a manuscript attempting to assess the development of BPH from molecular approaches and subsequently identify therapeutic targets.

Overall, the study was comprehensive, well written, and designed well. It uses cutting edge techniques and next-generation genomics technologies to examine the underlying molecular mechanisms of a specific type of benign prostatic hyperplasia. These data are needed in the benign urology field and this study provides data, resources, and new ideas for the research community. I recommend the authors to revise their manuscript to address specific concerns below before a final decision is reached.

1. A major criticism of this work that the authors present BPH as a single disease, enlarged prostate with glandular nodules. This is a very simplistic viewpoint and has hampered the field for decades. The authors need to state this emphatically in the manuscript as well as discuss it as a limitation of the study in the field of BPH. I am confident the authors are aware that BPH can be defined in many ways: enlarged prostate (big, bigger, and gigantic), symptomatic BPH, nodular (stromal vs glandular), smooth muscle contractility, fibrotic, and others. The BPH defined herein appears to be focused more so on the gigantic glandular hyperplasia; as such, it should be described in this manner. The description requested here will only help clear the "air" in the BPH field. The work presented here is very good and needs to see the light of day.
2. Another major criticism is the difficulty in determining the samples derivation (random, sequential, multiple studies, etc.). This is difficult to discern in part because the team has done an incredible amount of work and drew from many sources. Please clarify! Additionally, it is difficult to determine if all samples used were fresh or if some were archived FFPE or frozen or other. These details need to be reported in a succinct and easily comparable manner. It is critical for the reader to determine if the experiments are comparing "apples to apples" rather than "apples to oranges". This reviewer is less concerned with whether they were the perfect cohorts rather, if they are different that the reader can interpret why results are the way they are. An annotation table for the normal controls would be helpful to compare with the BPH patients. What were the Gleason or AUASI scores for the patients? The authors note that the patients are age matched. What was the age distribution? Since BPH is a disease of aging, a large majority of men without a diagnosis of BPH/LUTS still have histological changes associated with BPH alongside normal transition zone. Representative H&Es are necessary and would be helpful. Additionally, were the BPH patients on any form of medical therapy for BPH prior to surgery?
3. The identification of the different signatures was very interesting. Were all the nodules in the patients with the stromal signature predominantly glandular in nature? Were there stromal nodules? Representative histological images would be helpful to show no cellular differences between the two groups.
4. The WGS, WES, and SNP arrays bring a lot of interesting and important insight into BPH and conclude that BPH is not a neoplastic disease. This is consistent with the literature and has anecdotally been told so for the past several decades; thus the data presented here very nicely

demonstrate what the field has suspected for years, and the authors do so in a very good and thorough manner. However, in the introduction they do not build a convincing case that BPH may be construed as a neoplastic process. Please add a bit of verbiage to this cause.

5. As mentioned above, the authors need to be more specific about their definition of BPH and the type of BPH they are specifically examining. The authors are examining a small subset of BPH patients with extremely large prostates containing large/many nodules. In these cases, there might be limited/no normal adjacent normal; however, this is not always the case. Please remove or clarify the use of "normal TZ" from patients; clearly there is normal adjacent tissue in some BPH patients/specimens, but again perhaps not for these patients/samples.

6. The clustering to show that the BPH signature is specific to BPH and not to transition zone or peripheral zone was informative. What dataset was used to assess normal peripheral zone? This needs to be reported.

7. The patients treated with mTOR inhibitors, were they age matched to the BPH patient samples? Had these patients been diagnosed with BPH/LUTS previously? If mTOR inhibitors decrease prostate size regardless of genomic signatures in presumably normal prostates, would it be a drug that would be effective for the patients in BPH-B subgroups? Do these inhibitors have an effect on androgen production or AR signaling directly? These data are compelling and need to be discussed.

8. Figure S8 y-axis, is that gLand/stroma?

9. Figure 2, many of the axes and titles are symbols and not text.

10. Figure 3, a key for the heat maps in B and D.

11. Lines 131-133 suggest AR is downregulated in BPH. Numerous lines of evidence suggest AR/signaling is high, in fact 5aRIs are a cornerstone for the treatment of BPH. Please compare and contrast this as the authors results conflict here. Was AR and AR downstream targets (e.g. PSA, Nkx3.1) specifically looked at? Were circulating androgens evaluated?, were patients on drugs/diets that affected the androgen pathway?

12. Lines 161-163 state there was no clear enrichment of stromal cells visually. Does this mean that the molecular analysis was done on the tissue section or adjacent tissue that was fresh, frozen, other? Please clarify.

Below, the reviewer's comments are shown in black font, our response follows in **blue text**. New data in response to reviewers' comments is shown within each response, with the location of the data in the revised manuscript specified. If we have not included the presented data in the revised manuscript, we have explained our rationale, and labeled the figure presented here as "**For Reviewers.**" Any new text added to the manuscript is shown in *italics*.

Point-by-point Responses

Reviewers' comments:

Reviewer #1

In this study, Liu et al. sought the underlying molecular features of the transition to BPH via multi-modality molecular characterization of 18 cases. This is a well-performed study addressing an important question. The profiling is not particularly innovative, but comprehensive and the conclusions reasonable and well-written. While the molecular data prove largely negative (which is perhaps interesting in and of itself), the transcriptional information was information-rich and validate in part previously known facets of BPH. This is laudable attempt to answer an outstanding question in the urology/prostate cancer field. Findings are likely limited with respect to broader conclusions regarding benign precursor lesions in other settings, etc.

We thank the reviewer for their time and appreciate the careful review of the manuscript.

Major comments are:

1.) Regarding the cellularity of the specimens, can the authors highlight in greater detail their confidence in not having sequenced largely normal cells? Cellularity estimates are often discordant between pathology and analytical estimates from data, and this reviewer recognizes the complexity of doing so for non-tumor specimens in this case that largely lack the CNAs or molecular changes on which these estimates are typically based, but somatic mutations in largely diploid specimens (even if not drivers), may assist here? Such an analysis would be especially helpful in showing that the two RNA subgroups are different for reasons other than cellularity.

We understand the reviewer's interest and potential value of cellularity of the BPH samples, as a metric to understand and interpret the sequencing data.

One challenge in this disease state is that there is essentially no identifiable "adjacent normal" to use as a comparison – the entire anatomic region of the central prostate (transition zone) is often expanded/replaced by BPH. Therefore, our primary metric for defining the input material was pathologic assessment of the cored regions as histologically representing BPH.

As the reviewer suggested, we performed purity analysis on 18 BPH cases via using CLONET (PMID: 25160065; PMID: 31524989) based on the SNP array, WGS and WES data. However, as the reviewer expected, due to few copy number alterations and low mutation rates, the purity (and clonality) cannot be adequately calculated from BPH samples – only two samples showed any ability to calculate purity, and both we calculated as 50% “tumor” content, more consistent with artifact. The detailed results are available for the reviewer at:

https://drive.google.com/drive/folders/1TXFNuztoehRyvUPffg6lwHBet_djAgVE?usp=sharing

In addition, we ran ABSOLUTE (PMID: 22544022) on all samples to further confirm low “tumor” purity across all BPH samples, and there was no clear difference in calculated purity between the two molecular subgroups. These data are available for the reviewer at:

<https://drive.google.com/drive/folders/1Fl6vCbfOTzbRcUSuLFU3gKfdzkJUG1JO?usp=sharing>.

As suggested by the reviewer, we also explored interrogation of the diploid samples (which are most of the samples) – again, there were insufficient somatic alterations to return meaningful results.

Finally, we compared the allele frequency of somatic mutations between two subgroups, and found no difference between two subgroups based on all somatic mutations. As we expected, there were very few somatic mutations in BPH, and two RNA-defined subgroups showed similar allele frequency based on somatic mutations. These results are shown below:

FOR REVIEWERS: Boxplot of allele frequency between two BPH subgroups based on all somatic mutations from variant calling output.

We have not included these data in the manuscript given the limited utility to readers, but are happy to do so at the reviewer’s or editor’s discretion.

2. If the normal controls were from RPs of men with PCa but not BPH, were these samples

investigated alone and in detail in the molecular data to confirm they were normal prostate (no evidence molecular changes consistent with prostate cancers)? Such data would be helpful, its related to the interpretation of data throughout the manuscript, and would generally increase the confidence in the analysis.

We agree with the reviewer that the molecular data of normal controls would be helpful in confirming that these are indeed “normal” prostate tissue.

We have now examined the normal control samples for all known alterations in primary prostate cancer study (PMID: 26544944), and found no evidence of known somatic mutations or fusion genes in the normal controls (using the RNA-seq data). Furthermore, we see no evidence of gene expression changes consistent with prostate cancer in these control samples. These results are now included in supplementary table S7, figures S3 and S4.

We also altered the text in the methods section regarding control samples, adding:
“molecular analysis confirmed the absence of any cancer related molecular alterations”

We appreciate the reviewer’s thoughtfulness in suggesting these analyses, and feel that they have improved the manuscript.

3. Ostensibly, given the prevalence of BPH compared to primary prostate cancer (and the spectrum and frequency of drivers in the latter), it is unlikely to manifest clonal drivers per se, but what about evidence of subclonality in the data? Can the authors reasonably preclude the possibility of potential rare subclones (within reason vis a vis sensitivity) of neoplastic disease (perhaps in the cases with elevated PSA, or similar)? This is unlikely possible from the sequencing they have performed to date (due to the limited sensitivity of lower coverage broad sequencing), but perhaps they can examine the data for evidence of subclones at reasonably detectable frequencies?

We have now performed clonality analysis by CLONET (available for the reviewer at https://drive.google.com/drive/folders/1TXFNuztoehRyvUPffg6lwHBet_djAgVE?usp=sharing)

and by ABSOLUTE (available for the reviewer at <https://drive.google.com/drive/folders/1Fl6vCbFOTzbRcUSuLFU3gKfdzkJUG1JO?usp=sharing>),

However, due to the low frequency of genomic changes (as expected by the reviewer), we cannot identify the clonal or subclonal changes from BPH samples. To address this question with a different approach, we re-examined the our BPH WES and WGS data for all known somatic mutations in primary prostate cancer (PMID: 26544944), and we see no evidence of any of these genomic alterations, even at the lowest detectable frequencies. These results are now included in supplementary table S6.

We agree completely with the reviewer that this is an important but challenging question. Essentially, this is in the realm of “proving a negative”. We feel that we have made as good an effort as reasonably possible to find driver genomic alterations, although we certainly cannot rule out that our approach and the technology are not yet sufficient to do so. We certainly appreciate the reviewer’s thoughtfulness in suggesting these analyses, as we agree it is a key question.

We have altered the text in the discussion to reflect this possibility: *“While we can detect no somatic changes consistent with driver genomic alterations, given technological and sampling*

limitations, we cannot completely rule out the presence of genomic alterations below limits of detection.”

4. Is there absolutely no evidence of *ERG* fusion, even rarely, in the RNA? Would be a nice negative control to just show. Same goes for all known PCa drivers, of which the number is limited. A fill-out of sequencing reads at these loci in WGS/WES/RNA to show absolutely negative even at sub-detection thresholds would be nice to show.

We appreciate the reviewer’s suggestion, and agree that it does serve as a nice control. We indeed see absolutely no evidence of *ERG* fusion, with minimal/absent *ERG* expression (as expected in prostate) and no fusion reads in BPH cases from RNA-seq data. These results are now included in supplementary figures S3 and S4, and shown below.

We re-examined our BPH WES and WGS data for known somatic mutations in primary prostate cancer, and we see no evidence of any of these, even at the lowest detectable frequencies. These results are now included in supplementary table S6.

Figure S4. IGV snapshot of *TMPRSS2-ERG* fusion in prostate cancer samples with *TMPRSS2-ERG* fusion from TCGA study (PMID: 26544944), BPH and normal control samples.

5. Some images of the histology of cases (by stromal signal and not) would be appreciated, perhaps as supplementary materials to reaffirm the statement about the signature itself.

We thank the reviewer for this suggestion. We have now included the histological images of BPH cases in supplementary figure S1 (shown below).

Figure S1. Histological images of BPH cases from subgroups A and B, with percentage of epithelial cells shown at the bottom right.

6. The clinical data regarding mTOR inhibition is intriguing and a creative use of retrospective data, but as such, confounded by multiple factors that are difficult to control for in this kind of analysis (as the authors are well-aware). Moreover, the statistics here are exploratory and not consistent with proper univariate and multivariable outcome analyses. For those reasons, care should be taken in how strongly to articulate the conclusions of this analysis and the viability of mTOR inhibition as a therapeutic strategy for BPH without more data.

We thank the reviewer for the positive comments, and also agree completely with the reviewer that this analysis (by its very nature) is confounded by multiple factors, and should be interpreted with caution.

We have therefore added to the discussion:” *Despite this promising signal in clinical data, the conclusion that mTOR inhibitors can affect prostate size should be tempered by the limitations of this analysis. First, these are retrospective data with no ability to examine the molecular features of the responding patients. Future rigorously designed clinical studies, with the ability to stratify patients by molecular subgroup, will be necessary to validate and move these findings further toward clinical deployment.*”

In addition, in response to this concern as well as a similar one from Reviewer #2, we have altered the title of the manuscript to better reflect the exploratory nature of this data: “**Integrative genomic, transcriptomic, and epigenomic analyses of benign prostatic hyperplasia**”

7. In the absence of some degree of mechanistic confirmation (perhaps in organoids from BPH patients?) it's difficult to get a sense for validity of the mTOR association and some of the pathways suggested. This reviewer recognizes this is likely outside the scope of the present study and possible future work for the team, but its absence does limit the rigor and robustness of the conclusions and should likely be acknowledged in a conservative discussion of their findings.

We certainly agree with the reviewer that mechanistic confirmation of the *mTOR* association would increase confidence in the finding. However, the reviewer has astutely touched on another critical gap in the field – the lack of relevant BPH model systems in order to perform such mechanistic studies, and we are actively working to develop and credential a model system in which to study these pathways.

We agree with the reviewer that the lack of mechanistic validation should temper our conclusions. In addition to modifying the title and other changes above, we have added to the discussion: ”*additional mechanistic validation will be needed to clarify the nature of mTOR involvement and the specific signaling pathways in play.*”

Reviewer #2

In this well-written manuscript, the authors describe the molecular characterization of benign prostatic hyperplasia (BPH) using a variety of sequencing based approaches, including whole genome sequencing, transcriptome sequencing and methylation analysis. The cohort consists of 18 extreme cases of BPH as assessed by volume. Control samples from transitional zone and peripheral zone of normal prostate were used in some of the analyses. The study has generated a rather large dataset from BPH specimens and is the first whole genome sequencing description of this disease. While many of the findings in this regard are negative, (no structural rearrangements, low mutation burden lacking candidate drivers), this information is still important to our understanding of the biology of BPH. Results from methylation analyses and, to some degree, transcriptome analyses recapitulate previous studies. The inclusion of clinical data regarding prostate volume reduction following MTOR inhibition is intriguing, but is not linked directly to the molecular data generated by this study.

The manuscript is concise and does integrate multiple data types well. This reviewer has some concerns to address and further clarifications to request.

We thank the reviewer for their time and appreciate the positive comments.

Overall the data and the analytical results are presented very well. The conclusions follow the

data with a few exceptions mentioned next. These are the principal concerns. The categorization of the Subgroup B (BPH-B) is not clearly presented in the text or Figures. While it is different than Subgroup A in terms of the expression of genes coding for elements of metabolic pathways, it is not shown clearly to differ from controls. In fact, Subgroup A is shown to be different than controls for expression in metabolic pathways and then many of the following figures and supplement follow the theme that A is different than B, which, while true, does not support that B is different than controls. One could alternatively conclude that BPH-A differs from normal in both stromal signature and expression of metabolic pathways. An analysis and heat map representation including BPH-A, BPH-B and controls would help to clarify gene sets which might be useful to distinguish the 3 proposed classes.

We appreciate the reviewer's concern regarding the clarity of presentation for the BPH subgroups. We had originally shown the heatmap of all samples together in Figure 2B, but did not label the identified subgroups at this point for the sake of the narrative flow of the manuscript.

Therefore, we now have included the heatmap containing all samples (BPH-A, BPH-B and control samples) based on the BPH transcription signature derived from Figure 2B, after unsupervised clustering, as supplementary figure S5. We thank the reviewer for the suggestion to improve readability and interpretability,

The link between Subgroup A molecular profile and response to MTOR inhibition is not firmly established. This reviewer could not find any transcriptional characterizations of the responders or non-responders to MTOR inhibition. This can be understood, of course, as the MTOR clinical data is retrospective, without characterization of putative transcriptional classes. This reviewer does not know if archival biopsy material remains from those MTOR cases, but sequencing and analyses of such material would provide support for one of the major claims of the paper. Specifically, “found that MTOR inhibitors may be preferentially active in one subgroup” is not supported yet by the data in the paper - yet these conclusions are suggested in the title. The MTOR pathway has been known to regulate both organ and cell size in normal tissues

(PMID:12080086 and 22575479) and the responding cases in this paper have not been shown to have aberrant MTOR signaling.

We certainly agree with the reviewer that it would strengthen the link between BPH subtypes and clinical response to *mTOR* inhibition if we could assign subtypes to the patient cohort examined in Figure 4. Unfortunately, the nature of this data precludes this – this was a retrospective analysis of patient imaging and clinical data, without the ability to collect or analyze prostate tissue from these patients (these patients did not have prostate biopsies etc.). As such, this was meant to be only an exploratory analysis showing that *mTOR* inhibition, nominated by our molecular analyses, has real demonstrable clinical effects on the prostates of unselected patients. Unfortunately, we know of no cohort of prostate tissue available from patients that have been exposed to *mTOR* inhibitors and had imaging to capture prostate size changes.

We do agree with reviewer that the conclusion that Subgroup A specifically responds to *mTOR* inhibition should be tempered. We have therefore added to the discussion: *”Despite this promising signal in clinical data, the conclusion that mTOR inhibitors can affect prostate size should be tempered by the limitations of this analysis^{55, 56}. First, these are retrospective data with no ability to examine the molecular features of the responding patients. Future rigorously designed clinical studies, with the ability to stratify patients by molecular subgroup, will be necessary to validate and move these findings further toward clinical deployment. Second, additional mechanistic validation is will be needed to clarify the nature of mTOR involvement and the specific signaling pathways in play.”*

We have also changed the last sentence of the results section to soften our conclusion, and now reads: *“Overall, these data support that molecularly identified BPH subgroups represent biologically distinct subtypes of disease. Our analysis suggests one subgroup as dependent on mTOR signaling, and we show here in exploratory clinical analysis that mTOR inhibition affects prostate size, thereby nominating a potential novel therapeutic option.”*

In addition, in response to this concern, we have altered the title of the manuscript to better reflect the exploratory nature of this data: **“Integrative genomic, transcriptomic, and epigenomic analyses of benign prostatic hyperplasia”**

A lesser concern is the discrepancy between the stromal transcriptional signature and the lack of histological correlation. The authors do note this, but could some investigations be done to address this apparent discrepancy? Could the authors perform some IHC experiments to address the question could the stromal signature be due to a more “active” stroma, i.e is the expression of some of the signature genes in the stromal panel higher at the individual cell level in BPH vs normal?

We agree with reviewer’s concern about the discrepancy between the stromal transcriptional signature and the lack of histological correlation.

We explored the possibility of using IHC to define expression of specific stromal markers at an individual cell level. However, we concluded that this experimental approach would not be feasible: the absolute expression changes for any single gene are on the order of 1.5 fold, too low to be detectably different using a qualitative assay like IHC.

We therefore used an alternative approach to address this idea (also as suggested below by the reviewer). We examined gene enrichment analysis of multiple stromal-related signatures,

including those from Pollack et al. bulk RNA-seq and Henry et al. single-cell RNA-seq data. Interestingly, we found the distinct enrichment patterns of stromal signatures between the two BPH subgroups. Some – like the Pollack et al. stromal signature, and signal of stromal leukocytes are nearly identical between the subgroups. Others, however, are markedly different. These results are presented in supplementary figure S12, and are shown below.

In addition, we have added to the text of the results section:

“To gain further insight into the potential cellular components of these molecular data, we examined the signatures of different cellular compartments from recently reported single cell RNA-seq from normal prostate tissue (Figure S12)³⁹.”

Figure S12. Spider chart of normalized enrichment score of stromal signatures from Henry single-cell RNA-seq (PMID: 30566875) and Pollack bulk RNA-seq data (PMID: 31094703) in BPH subgroups. Different colors represent different subgroups when compared to normal control samples.

This provides some insight to the fact that, as the reviewer suggests, not all stroma is the “same”, and that differences in stroma that may be not be readily apparent to the eye of a pathologist. We thank the reviewer for this suggestion, as we feel it definitely adds to the manuscript (and allows us to leverage high quality publicly available data to do so). Certainly, further deconvolution of these stromal signature, and additional studies at single cell resolution, will contribute to our understanding of the pathogenesis of BPH and as disease classifiers. We hope that these data will help stimulate future studies in this direction for the field.

This reviewer commends the authors on their inclusion and analyses of relevant external datasets. Two lesser points- could the authors be more explicit that a stromal signature was previously found in BPH in the Pollack study. Second, could an attempt be made to analyze the proposed subgroup B vs the single cell data of Henry et al?

We understand the reviewer's interest about the stromal signatures from published studies. In the Pollack study (PMID: 31094703), they compared BPH patients with low versus high expression of a 65-gene stromal signature, comprising the core of the stromal gene feature.

As the reviewer suggested, we included the gene enrichment analysis of stromal signature from Pollack bulk RNA-seq and Henry single-cell RNA-seq data, and found the different enrichment of stromal signatures between two BPH subgroups. These results are presented in supplementary figure S13. In addition, we have modified the text of the results section, adding: "*consistent with the presence of stromal signatures in previous reports*" with the appropriate citation.

Minor points.

1. The age of men in the description of the frequency of BPH is missing (line 83)

We thank the reviewer for pointing out this oversight. We have made the corresponding changes in the text: "*The prevalence of BPH increases with age, with BPH symptoms reported by roughly 80% of men at age 70-79 (PMID: 22808960)*"

2. Principal component analysis is incorrectly called Principle component analysis at several places in the text and Figure 3.

We apologize for the incorrect spelling of "Principal component analysis," and thank the reviewer for pointing out the error. We have made the appropriate alterations throughout the text and figure legends.

3. A more explicit depiction of variant allele frequencies and perhaps clonality assessment would be informative to understanding the biology of BPH, could the authors add this?

We have included the variant allele frequencies in supplementary table S2. The detailed clonality results are available for the reviewer at:

CLONET

(https://drive.google.com/drive/folders/1TXFNuztoehRyvUPffg6IwHBet_djAgVE?usp=sharing) and ABSOLUTE

(<https://drive.google.com/drive/folders/1F16vCbFOTzbRcUSuLFU3gKfdzkJUG1JO?usp=sharing>).

We also compared the allele frequency of somatic mutations between two subgroups, and found no difference between two subgroups based on all somatic mutations, As we expected, there were very few somatic mutations in BPH, and two RNA subgroups showed similar allele frequency across based on somatic mutations. These results are shown below:

We have not included these data in the manuscript given the limited utility to readers, but are happy to do so at the reviewer's or editor's discretion.

4. The Figures are very nicely prepared-, there is a small problem with printing Figure 2, some of the captions and labels in the Figure print with a nonsensical font.

We apologize for labeling in Figure 2, and we have made the appropriate changes to the font to ensure compatibility.

Figure 2. BPH transcription and methylation profiles.

Reviewer #3 (Remarks to the Author):

The manuscript “Integrative genomic, transcriptomic, and epigenomic analyses of benign prostatic hyperplasia reveal new options for therapy” by Liu et al., is a manuscript attempting to assess the development of BPH from molecular approaches and subsequently identify therapeutic targets.

Overall, the study was comprehensive, well written, and designed well. It uses cutting edge techniques and next-generation genomics technologies to examine the underlying molecular mechanisms of a specific type of benign prostatic hyperplasia. These data are needed in the benign urology field and this study provides data, resources, and new ideas for the research community. I recommend the authors to revise their manuscript to address specific concerns below before a final decision is reached.

We thank the reviewer for their time and effort in critically evaluating the manuscript, and appreciate their positive comments.

1. A major criticism of this work that the authors present BPH as a single disease, enlarged prostate with glandular nodules. This is a very simplistic viewpoint and has hampered the field for decades. The authors need to state this emphatically in the manuscript as well as discuss it as a limitation of the study in the field of BPH. I am confident the authors are aware that BPH can be defined in many ways: enlarged prostate (big, bigger, and gigantic), symptomatic BPH, nodular (stromal vs glandular), smooth muscle contractility, fibrotic, and others. The BPH defined herein appears to be focused more so on the gigantic glandular hyperplasia; as such, it should be described in this manner. The description requested here will only help clear the “air” in the BPH field. The work presented here is very good and needs to see the light of day.

We thank the reviewer for pointing out that our presentation of the disease state is overly simplistic, and more is needed to explain the nuance of the field to the audience.

The characteristics of the BPH cases we profiled are defined in supplementary table S1, and were selected primarily on the basis of extremely large prostate size (top 1 percentile). We agree with the reviewer that these could be primarily described as gigantic glandular hyperplasia.

To try to guide the reader that these cases may not be completely generalizable to all, we had previously described in the introduction: *“We selected samples from patients with very large prostates (top 1 percentile and greater than 100cc, Table S1 and Figure 1A), based on the rationale that these “extreme outliers” were more likely to harbor biologically informative events^{27, 28}.”*

We have now added additional annotation to supplementary table S1, including prior medical therapy information. In addition, we have now added histologic images (Figure S1) for each sample to allow readers insight to the nature of each sample.

We have made the following changes to the text:

In the abstract, we added: *“from patients with very enlarged prostates.”* to the descriptions of cases, in order to highlight from the outset that our study may not be representative of all BPH.

In the discussion, we have added the following text to clarify the complexity of defining BPH in the field: *“However, BPH is a complex, heterogeneous, and pleomorphic disease that can be defined in many ways, likely representing a collection of distinct etiologies and pathobiologies^{3, 53, 54}.”*

Here, the BPH cases we interrogated were of “extreme” size (top 1 percentile of prostate size), which we hypothesized would be most likely to harbor informative molecular events, but may not be generalizable to more routine cohorts. “

Figure S1. Histological images of BPH cases from subgroups A and B, with percentage of epithelial cells shown at the bottom right.

2. Another major criticism is the difficulty in determining the samples derivation (random, sequential, multiple studies, etc.). This is difficult to discern in part because the team has done an incredible amount of work and drew from many sources. Please clarify! Additionally, it is difficult to determine if all samples used were fresh or if some were archived FFPE or frozen or other. These details need to be reported in a succinct and easily comparable manner. It is critical for the reader to determine if the experiments are comparing “apples to apples” rather than “apples to oranges”. This reviewer is less concerned with whether they were the perfect cohorts rather, if they are different that the reader can interpret why results are the way they are. An annotation table for the normal controls would be helpful to compare with the BPH patients. What were the Gleason or AUASI scores for the patients? The authors note that the patients are age matched. What was the age distribution? Since BPH is a disease of aging, a large majority of

men without a diagnosis of BPH/LUTS still have histological changes associated with BPH alongside normal transition zone. Representative H&Es are necessary and would be helpful. Additionally, were the BPH patients on any form of medical therapy for BPH prior to surgery?

We appreciate the reviewer's concerns regarding clarity of source material, and thank them for this suggestion. We agree completely that better explanation of the sample selection, annotation and histology of the BPH and normal control samples will improve the ability for readers to interpret the manuscript.

This was a single center study from Weill Cornell Medicine. These were not consecutive cases; patients with prostate size meeting inclusion criteria underwent informed consent for this IRB approved protocol prior to surgical therapy for BPH. All the BPH samples were derived from surgical specimens and prospectively collected and banked as frozen tissue. We have altered the methods section to incorporate these changes:

"This was a single center BPH study from Weill Cornell Medicine, and these were not consecutive cases. Patients with prostate size meeting inclusions criteria underwent informed consent for this IRB approved protocol prior to surgical therapy for BPH. All the BPH samples were derived from surgical specimens and prospectively collected and banked as frozen tissue."

We have now included annotation of the non-BPH controls in supplementary table S7, including age, Gleason score, percentage of cancer, preoperative PSA values and clinical stage information.

Regarding age matching, the mean age for BPH cases was 69, and ranged from 63 to 96, as shown in supplementary table S1. The mean age for control cases was 63. Our intent was not to imply these were "perfectly" age matched, but to reinforce that these samples were not from a grossly different patient cohort (e.g. young organ donors).

Representative H&E images of each BPH sample is now included in supplementary figure S1.

We have also now included prior medical therapy in supplementary table S1. As expected, all BPH patients were previously been treated with medical therapy before surgery, including 5-alpha reductase inhibitors.

We have now also discussed this in the text as a limitation to interpreting gene expression signaling involving androgen receptor signaling: *"However, as is common practice for BPH, all patients were exposed to medications affecting AR activity prior to surgery (5-alpha reductase inhibitors), making it unclear whether AR target gene changes were due to intrinsic properties of BPH or prior therapy."*

We appreciate the reviewer's thorough evaluation and suggestions to improve the reader's ability to understand the data, and feel that they have improved the manuscript.

3. The identification of the different signatures was very interesting. Were all the nodules in the patients with the stromal signature predominantly glandular in nature? Were there stromal nodules? Representative histological images would be helpful to show no cellular differences between the two groups.

We thank reviewer's suggestions about representative histological images. Therefore, we included representative H&E figures of BPH samples in supplementary figure S1, and shown below. In addition, we have annotated the percent epithelial content of each sample as

determined by GU pathologists. There is no discernable difference in epithelial content to pathologists between subgroups.

Figure S1. Histological images of BPH cases from subgroups A and B, with percentage of epithelial cells shown at the bottom right.

4. The WGS, WES, and SNP arrays bring a lot of interesting and important insight into BPH and conclude that BPH is not a neoplastic disease. This is consistent with the literature and has anecdotally been told so for the past several decades; thus the data presented here very nicely demonstrate what the field has suspected for years, and the authors do so in a very good and thorough manner. However, in the introduction they do not build a convincing case that BPH may be construed as a neoplastic process. Please add a bit of verbiage to this cause.

We apologize for not making this clearer: as the reviewer correctly inferred, we feel this is one of the more important conclusions of the study. We have added further language in the introduction to support hypothesis that BPH could represent a neoplastic process: *“Genomic driver alterations are identifiable in many benign neoplasms; for instance, uterine leiomyomas harbor recurrent MED12 mutations as well as complex chromosomal rearrangements (PMID:21868628, PMID:23738515), and profiling of hepatocellular adenomas has revealed multiple recurrent mutations (PMID:24735922).*

5. As mentioned above, the authors need to be more specific about their definition of BPH and the type of BPH they are specifically examining. The authors are examining a small subset of BPH patients with extremely large prostates containing large/many nodules. In these cases, there might be limited/no normal adjacent normal; however, this is not always the case. Please remove or clarify the use of “normal TZ” from patients; clearly there is normal adjacent tissue in some BPH patients/specimens, but again perhaps not for these patients/samples.

We apologize for the misleading annotations for the control samples in Figure 2.

We have made the corresponding changes in the text corresponding to Figure 2C and the legend of Figures 2A and 2C: *“Green color represents BPH samples, and grey color represents control samples from normal transition zones of men undergoing radical prostatectomy for prostate cancer (without clear evidence of BPH).”*

Figure 2. BPH transcription and methylation profiles.

6. The clustering to show that the BPH signature is specific to BPH and not to transition zone or peripheral zone was informative. What dataset was used to assess normal peripheral zone? This needs to be reported.

We tested BPH signatures on additional normal peripheral zone samples from previous RNA-seq studies (PMID: 25415230; PMID: 23555183; PMID: 22389870). Those frozen benign prostate tissues were collected as part of an Institutional Review Board–approved protocol and at time of radical prostatectomy at Weill Cornell Medicine.

We have included these references in the manuscript: “When compared to control samples from the normal peripheral zone (PMID: 25415230; PMID: 23555183; PMID: 22389870), this transcriptional signature was BPH specific, and not specific to transition zone tissue (Figure S5).”

7. The patients treated with mTOR inhibitors, were they age matched to the BPH patient samples? Had these patients been diagnosed with BPH/LUTS previously? If mTOR inhibitors decrease prostate size regardless of genomic signatures in presumably normal prostates, would it be a drug that would be effective for the patients in BPH-B subgroups? Do these inhibitors have an effect on androgen production or AR signaling directly? These data are compelling and need to be discussed.

We thank the reviewer for pointing out that this may need further explanation, and have now clarified our methodology. Subjects were not explicitly age matched to BPH samples, nor were they selected based on a preexisting history of BPH/LUTS. In this analysis, we used serial imaging data from male patients who were on an *mTOR* inhibitor for ANY reason in our institutional electronic medical record, and who had appropriate imaging data.

We have now clarified this point in our methods section: *“These patients were not selected based on a preexisting history of any specific disease state, symptoms, or age. The median age of patients at the time of initial CT scan was 58 years of age, and at follow up CT scan was 60 years of age.”*

In addition, we agree completely with the reviewer that these exploratory clinical data are very stimulating for the field regarding the predictors of response, and whether the patients with the best responses belonged to the expected subgroups. The retrospective and anonymized nature of this data precludes the ability to collect or analyze prostate tissue from these patients. As such, this was meant to be only an exploratory analysis showing that *mTOR* inhibition, nominated by our molecular analyses, has real demonstrable clinical effects on the prostates of unselected patients. Unfortunately, we know of no cohort of prostate tissue available from patients that have been exposed to *mTOR* inhibitors and had imaging to capture prostate size changes. We certainly believe this is worth exploring in future clinical and translational studies, and hope our study will provide impetus for many researchers in the field to do so.

We have also added additional text to the discussion to highlight potential mechanisms of action of *mTOR* inhibitors in the prostate: *“Second, additional mechanistic validation will be needed to clarify the nature of mTOR involvement and the specific signaling pathways in play. mTOR is a critical signaling pathway for nutrient sensing and cell growth, and has complex cross talk with other signaling pathways, including androgen receptor signaling.”*

8. Figure S8 y-axis, is that gland/stroma?

We apologize for the typo in the y-axis of Figure S8 (now it is Figure S11). It is gland/stroma, and we have corrected this.

9. Figure 2, many of the axes and titles are symbols and not text.

We apologize for wrong labeling in Figure 2, and we have made the corresponding changes in Figure 2 and made sure all fonts are broadly compatible.

10. Figure 3, a key for the heat maps in B and D.

We have added the key of clinical annotations for the heatmap Figures 3B and 3D.

Figure 3. Identification and validation of distinct BPH subgroups.

11. Lines 131-133 suggest AR is downregulated in BPH. Numerous lines of evidence suggest AR/signaling is high, in fact 5αRIs are a cornerstone for the treatment of BPH. Please compare and contrast this as the authors results conflict here. Was AR and AR downstream targets (e.g. PSA, Nkx3.1) specifically looked at? Were circulating androgens evaluated?, were patients on drugs/diets that affected the androgen pathway?

We appreciate the reviewer pointing out this discrepancy – we can certainly see the confusion this would cause, and apologize for not highlighting it previously.

As the reviewer correctly suspected, all the BPH cases were from patients that had been treated with 5αRIs (5-alpha-reductase inhibitors) prior to progressing to surgical therapy for BPH. This is a major confounder when interpreting this data – we cannot conclude whether these gene expression changes are due to effect of the prior therapy, or are innate to these BPH cases. We have now discussed this in the text as a limitation to interpreting the gene expression signaling involving androgen receptor signaling: “However, as is common practice for BPH, all patients were exposed to medications affecting AR activity prior to surgery (5-alpha reductase inhibitors),

making it unclear whether AR target gene changes were due to intrinsic properties of BPH or prior therapy.”

We have also now added the prior therapies for each patient to the clinical annotations of the cohort shown in supplementary table S1.

For this analysis, we used gene set enrichment on a commonly used composite set of androgen receptor target genes (PMID: 17010675; PMID: 26544944). In response to the reviewer’s comment about AR downstream targets, we have now shown expression levels of selected specific AR target genes in supplemental figure S7, shown below.

We did not have adequate sample remaining to evaluate circulating or tissue androgen levels.

12. Lines 161-163 state there was no clear enrichment of stromal cells visually. Does this mean that the molecular analysis was done on the tissue section or adjacent tissue that was fresh, frozen, other? Please clarify.

All DNA and RNA derived from BPH samples were extracted from frozen tissue blocks. Slides on each side of the block were cut, H&E stained, and annotated by an expert GU pathologist for areas to be cored from the block.

To clarify the workflow, we have added in the methods: *“Patients with BPH (benign prostatic hyperplasia) were prospectively enrolled for sequencing of prostate frozen tissue samples from transitional zones under a protocol approved by the institutional review board of Weill Cornell Medical College. Slides were cut from frozen blocks, stained with hematoxylin and eosin, and*

areas for coring were designated. Slides were annotated for epithelial and stromal content by expert GU pathologists.”

Reviewers' Comments:

Reviewer #1:

Remarks to the Author:

The authors have addressed my previously articulated concerns and revised the manuscript accordingly.

Reviewer #2:

Remarks to the Author:

This reviewer thanks the authors for their responses to all reviewers concerns. The responses are sufficient, and aside from experiments that could not be done due to retrospective nature of the study, are complete. I can recommend publication of the current manuscript without reservation.

Reviewer #3:

Remarks to the Author:

The authors have employed a comprehensive molecular investigation of BPH (genomic, transcriptomic and epigenetic) in 18 BPH cases from patients with very enlarged prostates. Their claims of non-neoplastic progression are consistent with the literature; moreover their data is supportive of this claim. Thus, for this type of BPH their claim is substantiated. The authors also identify two types of BPH as determined by their analyses, even though histological evidence is lacking. Again, based on the molecular analyses, this claim is substantiated. Lastly, the authors claim that mTOR pathways are involved in at least one type of BPH identified by these studies.

The readers will benefit from this research in part due to validation of what the literature has supported for decades, that BPH is an adenoma rather than a malignancy. Furthermore, the larger community will likely appreciate the difficulty of treating BPH as it is a complex disease that is not that different than "normal". Lastly, data shown herein give hope to those suffering from this disease, by shedding light on a new therapeutic target, mTOR pathway, for treatment or prevention.

This work should help shape the field of benign urology. It provides a clearer picture to this type of BPH and BPH as a whole. Further it provides a rich resource for other researchers to mine and inform future research.

The reviewers adequately address the reviewers concerns and the manuscript is stronger now than in the prior submission. Sufficient for publication.